# Osage Orange (*Maclura pomifera*) and Spearmint (*Mentha spicata*) Leaf Extracts Exhibit Antibacterial Activity and Inhibit Human Respiratory Syncytial Virus (hRSV)

**DOI:** 10.3390/pathogens14080776

**Published:** 2025-08-05

**Authors:** Milica Nenadovich, Molly Kubal, Maci R. Hopp, Abigail D. Crawford, Megan E. Hardewig, Madison G. Sedlock, Rida Jawad, Zarrar A. Khan, Adrianna M. Smith, Mia A. Mroueh, Matthew DuBrava, Ellie C. Jones, Cael Rahe, Sean T. Berthrong, Anne M. Wilson, Michael P. Trombley, Ashlee H. Tietje, Christopher C. Stobart

**Affiliations:** 1Department of Biological Sciences, Butler University, Indianapolis, IN 46208, USArjawad@butler.edu (R.J.); mrouehmiaa@gmail.com (M.A.M.); ecjones3@butler.edu (E.C.J.); crahe@butler.edu (C.R.);; 2Clowes Department of Chemistry and Biochemistry, Butler University, Indianapolis, IN 46208, USA

**Keywords:** antiviral, cancer, phytochemistry, respiratory syncytial virus, antibacterial

## Abstract

The increasing prevalence of antibiotic resistance and the limited availability of antiviral therapeutics for pathogens such as human respiratory syncytial virus (hRSV) underscore the need for novel, plant-derived antimicrobial substances. In this study, we evaluated the antiproliferative, antibacterial, and antiviral activities of aqueous leaf extracts from two plants commonly found in North America, Osage orange (*M. pomifera*) and spearmint (*M. spicata*). Both extracts exhibited no significant cytotoxic or morphologic impact on HEp-2 human cancer cells up to 25 mg/mL. However, both extracts demonstrated strong dose-dependent antibacterial activity, significantly inhibiting replication of *E. coli* and *S. aureus* at concentrations ≥ 1 mg/mL. Antiviral assays revealed that both extracts inhibited hRSV infectivity, with spearmint extract showing higher potency (EC_50_ = 1.01 mg/mL) compared to Osage orange (EC_50_ = 3.85 mg/mL). Gas chromatography–mass spectrometry (GC-MS) identified three major extract constituents: 3-hydroxybenzyl alcohol, 4-hydroxybenzyl alcohol (Osage orange), and R-(-)-carvone (spearmint). Among these, only carvone significantly inhibited hRSV in vitro, suggesting its key role in spearmint’s antiviral activity. These findings highlight the therapeutic potential of Osage orange and spearmint leaf extracts, particularly as sources of water-soluble compounds with antimicrobial properties, and support further investigation into their mechanisms of action and broader clinical relevance.

## 1. Introduction

Plant extracts, essential oils, and phytochemicals have been used in traditional medicine throughout antiquity to treat a wide range of infections and ailments. Today, according to the WHO, it is estimated that 80% of the world still relies on plant-based therapies, and 40% of all modern pharmaceutical products can be traced to plants [1]. Some of the greatest challenges facing modern medicine include increased incidence of cancer, proliferation of antimicrobial resistance, and emerging infectious diseases [2,3,4,5,6,7,8]. These challenges demand the continued investigation of novel therapies and approaches.

In December 2019, a novel coronavirus (SARS-CoV-2) emerged and triggered the global COVID-19 pandemic [9]. The lack of initial vaccine and therapeutic options led to increased morbidity and mortality early during the pandemic response. The eventual development, approval, and implementation of nirmatrelvir in November 2021 was instrumental in changing outcomes for severe cases of COVID-19 [10,11]. Like SARS-CoV-2, there remain many viruses for which we continue to lack therapeutic options. Human respiratory syncytial virus (hRSV) is a major upper and lower respiratory pathogen responsible for high rates of hospitalizations of infants and the elderly [12]. Despite significant advances of recent in vaccine development, there remain no commercially available therapeutic options for hRSV.

In this study, we investigate the antiproliferative, antibacterial, and antiviral potential of two plants commonly found in North America: Osage orange and spearmint. Osage orange (*Maclura pomifera*) is a deciduous tree of the mulberry family (*Moraceae*) originally native to the south central United States and is notable for its large yellow-green fruit often called “hedge apples” [13,14]. Today, Osage orange can be found throughout much of North America. The wood of Osage orange withstands rot remarkably well and is incredibly dense, making it great for use in fence posts, bows, and tool handles [15,16]. The yellow coloration of the wood has also led to its use as a dye [17]. Lastly, compounds present in the plant have also been extracted for use as insect repellent as well as an ointment to treat sore eyes [18,19,20]. Due in large part to its widespread uses, several studies have evaluated Osage for antimicrobial properties; however, most of these studies have focused on the antibacterial properties of the wood and fruit of the plant rather than leaves of the plant [14]. One previous study looking at Osage orange leaf extracts demonstrated broad inhibitory activity against a diverse number of bacterial species [21]. However, little remains known of the antiviral activity of Osage orange leaf extracts, specifically against hRSV.

Spearmint (*Mentha spicata*) is a perennial herb which originates in Europe and Asia but has now been spread to nearly all temperate regions of the world [22]. It has been widely used for thousands of years to treat an impressive array of diseases and disorders including respiratory disease, diabetes, headaches, pain, and stomach and digestive disorders [23]. Extracts and essential oils have been prepared from many different parts of the plant [22,23]. Chemical analysis of spearmint extracts and oils have identified numerous key components which include carvone, carvacrol, limonene, 1, 8-cineole, menthone, and linalool [23,24,25,26]. Of these components, carvone has been widely shown to be the primary constituent present in spearmint essential oils and is principally responsible for its distinct odor. Previous investigations have extensively studied the biological activities of spearmint extracts and essential oils, which include antiproliferative and antioxidant effects on cancer cells and antimicrobial properties towards a wide range of bacteria, fungi, viruses, and parasites [23,26,27,28,29]. However, there are inconsistent results across many of these studies and its activity towards the human respiratory pathogen human respiratory syncytial virus (hRSV) is unknown [23].

## 2. Materials and Methods

### 2.1. Osage Orange and Spearmint Extract Preparation

Osage orange (*M. pomifera*) and spearmint (*M. spicata*) leaves were collected on the campus of Butler University, a private, liberal arts university located in Indianapolis, IN USA (39°50′19.85″ N, 86°10′21.22″ W). Leaves were initially identified and cataloged within the Friesner Herbarium at Butler University (accession numbers 20240116 and 20240121, respectively) and can be examined upon request. Leaves were initially separated from stems and deveined prior to being dried in a convection oven at a temperature of 32 °C (90 °F). The dried leaves were then subsequently coarsely chopped, weighed, and then suspended in deionized water for 5 min at 95 °C to produce an extract concentration of 100 mg/mL. The extract was initially passed through a coffee filter to remove insoluble particulates and then aliquoted and stored until use at −80 °C. Immediately prior to experimentation, frozen aliquots were thawed and filter-sterilized through a 0.2-micron filter to render them free of microbial contaminants.

### 2.2. Mammalian Cell Culture

HEp-2 cells, a HeLa-derived human cancer cell line (ATCC CCL-23) were cultured in complete Dulbecco’s Modified Eagle Medium (DMEM; Corning Inc., Manassas, VA, USA) supplemented with 5% fetal bovine serum (FBS; Rocky Mountain Biologicals LLC, Missoula, MT, USA) and an antimicrobial mixture (Corning Inc., Manassas, VA, USA) containing 50 μg/mL penicillin, 50 μg/mL streptomycin, and 2.5 μg/mL amphotericin B [30]. The cells were maintained at 37 °C under 5% CO_2_.

### 2.3. GC-MS Chemical Analysis of Extract Constituents

Plant extracts were analyzed by gas chromatography–mass spectrometry (GC-MS) after organic extraction following a procedure previously described [31,32]. In brief, 1 mL of each plant extract was combined with 10 mL of methanol and extracted in 500 mL of ethyl acetate and 200 mg of anhydrous NaCl. The organic phase of the extraction was separated and analyzed using a ThermoFisher Scientific TRACE 1310 GC-ISQ LT gas chromatography-mass spectrometry (GC-MS) instrument (Thermo Scientific, Waltham, MA, USA). The instrument was run using Xcalibur (v4.5), and data were processed using FreeStyle 1.8. Major constituents identified from the assay were identified using the NIST compound library accessed through Compound Discover.

Major peaks were analyzed and compared to the library for identification. Those which had a high probability of being accurately identified were ordered. The compounds identified and assessed in this report were 3-hydroxybenzyl alcohol (Ambeed Inc., CAS 620-24-6, Arlington Hts, IL USA), 4-hydroxybenzyl alcohol (Ambeed Inc., CAS 623-05-2, Arlington Hts, IL, USA), and R-(-)-carvone (Ambeed Inc., CAS 6485-40-1, Arlington Hts, IL, USA). The relative concentrations of these compounds in the parent extract were determined by comparison to a loaded methanol control. Solutions of each constituent were prepared in DMSO and stored according to manufacturer recommended storage conditions.

### 2.4. MTS Cell Viability Assays

A colorimetric assay using 3-(4,5-dimethylthiazol-2-yl)-5-(3-carboxymethoxyphenyl)-2-(4-sulfophenyl)-2H-tetrazolium (MTS) was performed using the CellTiter 96 Aqueous One Solution Cell Proliferation Assay as described by the manufacturer (Promega Corp., Madison, WI, USA) on HEp-2 cells incubated for 24 h with varying doses of either plant extract or chemical. The percent cell viability was determined as the ratio of absorbance (after background subtraction) at 490 nm of treatment compared to untreated control.

### 2.5. Immunofluorescence Microscopy

HEp-2 cells were cultured on glass microscope slide coverslips in 12-well plates at a density of 10^5^ cells per well for 24 h at 37 °C under 5% CO_2_. After attachment, the cells were incubated with varying doses of either plant extract or chemical for an additional 24 h. To prepare the cells for visualization by immunofluorescence (IF), the cells were fixed, permeabilized, and stained using FITC-conjugated anti-tubulin antibody (1:200 dilution), TRITC-conjugated Phalloidin (1:300 dilution), and DAPI (4′,6-diamidino-2-phylindole; 1 μg/mL). IF images were acquired using a Leica DM5500 fluorescence microscope. Nuclei counts were performed on six images collected from two cover slips to quantify the percent of mitotic and apoptotic nuclei after treatment. Cytoskeleton scores for disruption of actin microfilaments and tubulin microtubules were performed using a Likert scale ranging from 1 to 4 for normal cytoskeleton formation to complete cytoskeletal collapse. The identity of treatment was blinded during scoring.

### 2.6. Bacterial Replication Inhibition Assay

Antibacterial activity was initially performed using cultures (Avantor Inc., Radnor Twp, PA, USA) of *Escherichia coli* and *Staphylococcus aureus* cultured in Tryptic Soy Broth (TSB) under shaking at 37 °C. TSB solutions were co-inoculated with varying doses of either plant extract or chemical and aliquots of either *E. coli* or *S. aureus* at mid-log growth. Culture growth was assessed at 0, 1, 3, and 5 h by optical density at a wavelength of 600 nm. The fold increase in growth was determined as a ratio of the absorbance at each time point compared to the initial absorbance of the solution.

### 2.7. Bacterial Disk Diffusion Assays

A disk diffusion experiment was performed in triplicate by adding 20 μL of undiluted (100 mg/mL), 10% (10 mg/mL), 1% (1 mg/mL), 0.1% (0.1 mg/mL), or 0% of each extract to sterile, blank 6 mm disks. The disks were placed on Mueller–Hinton agar (MHA) plates which were inoculated for lawn growth with cultures of either *E. coli* or *S. aureus.* The plates were incubated at 37 °C for 24 h before the zone of inhibition was measured.

### 2.8. Viral Inactivation and Replication Assays

A previously described recombinant strain (A2-mKate2) of human respiratory syncytial virus (hRSV), which expresses a far-red fluorescent reporter (monomeric Katushka 2) was used to evaluate antiviral activity [33]. Viral inhibition assays were performed by inoculating HEp-2 cells cultured at 37 °C under 5% CO_2_ with complete DMEM containing varying doses of either extract or chemical and hRSV A2-mKate2 at a multiplicity of infection (MOI) concentration of 0.05 infectious particles (FFU) per cell. The assay was allowed to incubate for 24 h prior to visualization using a Leica DMIL phase-contrast microscope with a Texas Red filter to quantify the number of infected cell foci. The percent reduction in virus was calculated as the ratio of the number of infected foci after treatment compared to the number of foci after no treatment. These data were used to calculate the effective concentration, which reduces the virus by 50% (EC_50_). Viral replication assays were performed by infecting HEp-2 cells at a multiplicity of infection (MOI) of 0.05 infectious fluorescent focus units (FFU) per cell with hRSV A2-mKate2 in the presence of different doses of plant extracts. Aliquots of supernatant were obtained and frozen at 0, 1, 2, and 3 days post-infection and were subsequently thawed and titered as previously described [34]. The average titer was determined from three experimental replicates.

### 2.9. Statistical Analyses

Statistical analyses were performed on all quantified experiments using RStudio (ver. 3.6.0) and a CAR package (ver. 3.1-3). Analyses of covariance (ANCOVA) were used to identify significance compared to control treatments during cytotoxicity, antibacterial, and antiviral assays. IF microscopy data were analyzed using an analysis of variance (ANOVA) with post-hoc multiple comparisons (Tukey’s HSD) when comparing treatments to controls.

## 3. Results

### 3.1. Analysis of Impacts of Osage Orange (M. pomifera) and Spearmint (M. spicata) Leaf Extracts on Mammalian Cancer Cell Viability and Morphology

To assess the activity of Osage orange and spearmint leaf extracts on mammalian cells, HEp-2 cells (which are both susceptible and permissive for hRSV infection), were initially evaluated for cytotoxicity using an MTS-based assay after exposure to varying doses of extract (Figure 1A). No significant cytotoxicity was observed for either extract up to doses of 25 mg/mL. However, there was a notable significant increase (*p* = 0.001) in cell viability observed at nearly all concentrations of spearmint tested. These data indicate that neither Osage orange nor spearmint extracts promote cell death in HEp-2 cells; however, it remained unclear if there was any perturbation in structure or replication induced by exposure to the extracts.

Immunofluorescence microscopy was used to evaluate both changes in nuclear morphology as well as the cell cytoskeleton. Images were obtained of cells exposed to 10 mg/mL of each extract (Figure 1B). Compared to the water control, there were similar levels of cell density observed after treatment with either extract. However, there was a notable decline in cell density after exposure to the known pro-apoptotic control, colchicine. In addition, there were some changes in cytoskeleton observed across treatments. Multiple images were obtained of each treatment to quantify differences in nuclear and cytoskeletal morphology (Figure 1C). Consistent with its role as an inducer of apoptosis, colchicine resulted in a significantly higher average of 19% of cells exhibiting altered nuclei compared to the average 1% observed after water treatment (*p* = 0.00032). There were no significant differences observed for apoptotic nuclei between either extract treatment and water. In addition to quantifying nuclei for apoptosis, nuclei were also identified for indications of active mitosis. No significant differences in mitotic nuclei were observed between any of the treatments. Lastly, blinded scoring of cytoskeletal disruption was performed between treatments. Colchicine treatment resulted in significant disruption of the microfilament and microtubule cytoskeleton compared to water treatment (*p* = 0.006). However, there were no notable differences in cytoskeletal morphology between either extract treatment and the water control. Collectively, these data show treatment of HEp-2 cells with either extract did not result in significant cytotoxicity or altered morphology. Furthermore, these data indicate that any observed impacts on hRSV viral replication to be evaluated later are not likely due to disruption of cell structure or function.

### 3.2. Evaluation of Antibacterial Activity of Osage Orange (M. pomifera) and Spearmint (M. spicata) Leaf Extracts

Osage orange and spearmint extracts were next evaluated for antibacterial activity using cultures of *E. coli* and *S. aureus* (Figure 2). Both extracts promoted significant dose-dependent reductions in *E. coli* replication (Figure 2A). Osage orange extract reduced *E. coli* replication relative to no treatment at concentrations of 5 mg/mL (*p* < 0.001) and 1 mg/mL (*p* = 0.005). Spearmint extracts significantly reduced *E. coli* replication at both 5 mg/mL (*p* = 0.002) and 1 mg/mL (*p* = 0.004) as well as the lower concentration of 0.1 mg/mL (*p* = 0.02). At the greatest concentration tested of 5 mg/mL, Osage orange reduced the fold increase in OD_600_ to 10.2% of the growth observed in the untreated control after 5 h. The reduction after treatment with spearmint extract was more profound with a fold increase of only 3.2% of the growth observed in the untreated control. Similar to treatment of *E. coli*, both Osage orange and spearmint extracts also exhibited significant dose-dependent antibacterial activity against S. aureus at doses of 5 mg/mL (both *p* < 0.001) and 1 mg/mL (both *p* < 0.001). At 5 mg/mL, the fold increase in OD_600_ was reduced to 15.7% and 10.2% for Osage orange and spearmint extracts relative to untreated control.

A subsequent experiment was performed to identify the zone of inhibition associated with varying doses of Osage orange and spearmint extracts (Figure 2B). Disks were infused with doses ranging from undiluted (100 mg/mL) to no extract and were incubated on plates inoculated with either *E. coli* or *S. aureus*. At a concentration of 100 mg/mL, both extracts resulted in small zones of inhibition against each bacterial strain. Osage orange extract at 100 mg/mL resulted in an average zones of inhibition of 6.83 ± 0.29 mm and 6.50 ± 0.50 mm, respectively, against *E. coli* and *S. aureus*. Spearmint extract induced larger observed zones of 7.83 ± 0.76 mm and 7.17 ± 0.58 mm against *E. coli* and *S. aureus*, respectively. Upon exposure of *E. coli* to a 10% solution (10 mg/mL) of each extract, zones of inhibition of 6.17 ± 0.29 mm and 6.67 ± 0.50 mm were measured for Osage orange and spearmint, respectively. Additionally, a measurable zone of inhibition (6.50 ± 0.87 mm) was detected for a 10 mg/mL spearmint extract against *S. aureus*. However, no zones were observed around disks treated with 10 mg/mL of Osage orange extract against *S. aureus* and for neither extract at doses of 1 mg/mL or below.

Overall, these data demonstrate that at concentrations of at least 1 mg/mL, both Osage orange and spearmint extracts harbor significant antibacterial activity against the replication of two common species of bacteria. Testing of inhibitory activity by disk-diffusion shows complete inhibitory activity of both extracts at a concentration of 100 mg/mL and varying levels of activity at 10 mg/mL.

### 3.3. Antiviral Activity of Osage Orange (M. pomifera) and Spearmint (M. spicata) Leaf Extracts Against Human Respiratory Syncytial Virus (hRSV)

To evaluate the inhibitory activity of Osage orange and spearmint leaf extracts on infectivity of human respiratory syncytial virus (hRSV), HEp-2 cells were inoculated with a recombinant hRSV strain A2 expressing a red fluorescent reporter protein in the presence of varying concentrations of extract (Figure 3A). Extracts were tested at concentrations up to 2.5 mg/mL. Previous work has shown that hRSV is potently inhibited by common sage (Salvia lyrata) [32]. Sage was used as a control in this experiment and resulted in significant inhibition with dose-dependent reductions in virus as previously observed. Treatment with Spearmint extract resulted in a similar significant dose-dependent reduction (*p* < 0.001) in hRSV to the sage treatment control with an effective concentration to reduce viral infectivity by 50% (EC_50_) of 1.01 ± 0.07 mg/mL (compared to 1.15 ± 0.13 mg/mL for sage). At the highest concentration tested (2.5 mg/mL), spearmint extract reduced hRSV infectivity to 22% of the amount observed in the untreated control. While still significant (*p* = 0.009), treatment with Osage orange was less effective at inhibiting hRSV. The EC_50_ for Osage orange was found to be 3.85 ± 1.91 mg/mL with hRSV infectivity reduced to 62% of the infectivity in the untreated control at the 2.5 mg/mL dose. These data indicate that both Osage orange and spearmint inhibit hRSV infectivity albeit at different levels of efficacy.

In order to identify the impacts of these extracts on hRSV replication, viral growth curves were performed by determining virus yield over 3 days of infection (Figure 3B). HEp-2 cells were infected with a multiplicity of infection (MOI) of 0.05 virus particles per cell in the presence of extract concentrations of 0.1 mg/mL and 1 mg/mL. Compared to untreated (0 mg/mL) controls, notable differences in virus replication as determined by mKate2 reporter expression observed visually at 2 d post-infection (d p.i.) during treatment with 1 mg/mL of either spearmint or sage extract. However, no apparent differences in signal were observed at 0.1 mg/mL nor for any dose of treatment for Osage orange. To quantify the amount of virus released, the supernatant was obtained at each time point and the amount of virus quantified by FFU titering. The supernatant (rather than whole-cell monolayer scraping) was used for virus calculation to ensure that virus was released and had completed the replication cycle. Consistent with the greatest differences being observed in replication at 2 d p.i., the average titer of the supernatant during spearmint treatment at a dose of 1 mg/mL was 2.2 (±0.4) × 10^1^ FFU/mL compared to 6.2 (±1.6) × 10^2^ FFU/mL, a significant 28-fold reduction in titer. Treatment with 1 mg/mL of sage (which had previously been reported to inhibit hRSV) resulted in an average 31-fold reduction in titers compared to the untreated control. No differences in average titer were observed at 2 d p.i. after treatment with Osage orange at either dose or during treatment with any of the extracts at the lower concentration of 0.1 mg/mL. By 3 d p.i., the cell monolayer had begun to dissociate in wells with more advanced infection. There were no significant difference in detected titers between extract treatments and the untreated control by 3 d p.i. These data show that spearmint extract was able to separately inhibit hRSV infectivity and virus replication at concentrations of at least 1 mg/mL. While Osage orange extracts show more modest inhibitory activity on hRSV infectivity, no significant reductions in replication were observed up to concentrations of 1 mg/mL.

### 3.4. Characterization of the Major Chemical Constituents Found in Osage Orange (M. pomifera) and Spearmint (M. spicata) Leaf Extracts on HEp-2 Cancer Cells

To identify some of the principal compounds present in Osage orange and spearmint which may be responsible for its antimicrobial activity, the extracts were analyzed by GC-MS (Figure 4A). Two major constituents were identified in Osage orange extract (3-hydroxylbenzyl alcohol and 4-hydroxybenzyl alcohol) and one in spearmint extract (R-(-)-carvone. All three of these compounds (3- and 4-hydroxybenzyl alcohol and R-(-)-carvone) were commercially available and were obtained to test for biologic activity. The approximate concentrations of these chemical constituents were determined based on normalization to a spiked methanol control in the GC-MS. The samples were prepared in DMSO and first evaluated for cytotoxicity in HEp-2 cancer cells at concentrations up to the approximate concentrations estimated from the GC-MS results (Figure 4A). No significant reductions in cytotoxicity were observed for 3- and 4-hydroxybenzyl alcohol up to concentrations of 39 mM. R-(-)-carvone treatment had minimal effect at most lower concentrations but did exhibit a dose-dependent cytotoxic response which was significant (*p* = 0.00004) at concentrations leading up to 17 mM near the aqueous saturation limit of solution.

To determine any impact these chemicals had on nuclear and cytoskeletal morphology, immunofluorescence images were acquired of HEp-2 cells treated with each of the chemicals (Figure 4B). Visually, there were no notable changes in cell density; however, there appeared to be some alteration in cytoskeleton in response to several treatments. To quantify any possible differences, images were assessed for nuclear alteration and cytoskeletal disruption (Figure 4C). Compared to a DMSO control, there were significant reductions in the number of mitotic nuclei observed after treatment with both 3-hydroxybenzyl alcohol (*p* = 0.046) and 4-hydroxybenzyl alcohol (*p* = 0.008) in addition to the colchicine control (*p* = 0.005). No significant difference in mitotic nuclei was observed after treatment with R-(-)-carvone. Quantification of altered nuclei consistent with apoptosis found only the colchicine control inducing significantly greater apoptosis (*p* = 0.002) compared to the DMSO control.

Lastly, the extent of cytoskeleton disruption was quantified between treatments and both 3-hydroxybenzyl alcohol (*p* = 0.003) and carvone (*p* = 0.0005) (in addition to the colchicine control) promoted increased cytoskeletal disruption relative to the DMSO control. In summary, 3-hydroxybenzyl alcohol and 4-hydroxybenzyl alcohol did not promote any significant change in cytotoxicity but was associated with a significant reduction in mitotic nuclei and increased cytoskeletal morphology. R-(-)-carvone treatment resulted in significant reductions in cell viability at the highest concentration tested and induced increased cytoskeleton disruption compared to the DMSO control.

### 3.5. Antibacterial and Antiviral Activity Associated with Chemical Constituents Associated with Osage Orange (M. pomifera) and Spearmint (M. spicata) Leaf Extracts

3- and 4-hydroxybenzyl alcohol and R-(-)-carvone were next evaluated for antibacterial activity against cultures of *E. coli* and *S. aureus* (Figure 5A). Treatment of *E. coli* with 4-hydroxybenzyl alcohol (*p* = 0.0057) and R-(-)-carvone (*p* = 0.039) resulted in significant reductions in replication at the highest tested concentrations of 390 μM and 170 μM, respectively. A similar trend was observed in treated cultures of *S. aureus*. Both 4-hydroxybenzyl alcohol (*p* = 0.0057) and R-(-)-carvone (*p* = 0.039) inhibited *S. aureus* replication at the 390 μM (*p* = 0.00027) and 170 μM (*p* = 0.019) concentrations, respectively. However, 4-hydroxybenzyl alcohol, was also capable of significantly reducing *S. aureus* replication at the 10-fold lower concentration of 39 μM (*p* = 0.00002). Replication of both *E. coli* and *S. aureus* remained largely unaffected at all tested concentrations of 3-hydroxybenzyl alcohol. Lastly, the activity of 3- and 4-hydroxybenzyl alcohol and R-(-)-carvone were tested against the human respiratory pathogen hRSV (Figure 5B). Co-treatment of HEp-2 cells with either 3- or 4-hydroxybenzyl alcohol and hRSV failed to reduce viral infectivity at all tested concentrations up to the approximate concentration found in the parent Osage orange extract. However, a significant dose-dependent decline in hRSV infectivity was observed with treatment of R-(-)-carvone (*p* = 0.003). The EC_50_ for R-(-)-carvone was found to be 3.09 ± 0.34 mM.

## 4. Discussion

In this study, we evaluated the biologic activity of leaf extracts prepared from Osage orange (*M. pomifera*) and spearmint (*M. spicata*), two common plants found in North America. Our study reports that aqueous extracts prepared from the leaves of Osage orange and spearmint exhibit the ability to significantly inhibit two species of bacteria and the major respiratory pathogen, hRSV, with little effect on the viability or cell morphology of the host HEp-2 mammalian cancer cells.

Prior to this study, little was reported regarding the antiproliferative effects of Osage orange leaf extracts. Earlier studies have shown that essential oils and extracts from bark and fruit of the plant exhibit antioxidant and antiproliferative activities [14,35,36]. We show here that Osage orange aqueous leaf extracts did not alter the growth or cell morphology of a HEp-2 cancer cell line. There are differing reports regarding whether spearmint extracts and essential oils harbor cytotoxic and pro-apoptotic activity against cancer cells and the nature of how it might promote antiproliferative effects [23,37]. Furthermore, several studies have indicated that treatment of cancer cells directly with the primary chemical constituent of spearmint, carvone, results in cytotoxic and pro-apoptotic effects [38,39]. We show in this study that at concentrations of 10 mg/mL, spearmint whole leaf extracts had no significant reduction in cell viability, induction of apoptosis, or cytoskeletal alterations. However, use of carvone at tested higher concentrations was found to promote cytotoxicity and induce cytoskeletal disruptions. While these data may appear to contradict one another, it has been reported that the concentration of carvone can vary based on geographic location and tissue source in the spearmint [23]. It is important to acknowledge that a limitation of this study is that our findings are limited to a single cell line, these data may suggest that the antiproliferative and pro-apoptotic activity of spearmint extracts are cell or tissue specific rather than of a general nature.

Consistent with previously reported studies, we found that both Osage orange and spearmint were able to inhibit the replication of two species of bacteria (*E. coli* and *S. aureus*; Figure 2). However, our studies have specifically shown that these activities are also retained within the aqueous extracts of the leaves at concentrations as little as 1 mg/mL of the prepared extract. In the only study (best to our knowledge) which previously reported on antibacterial activity of specifically leaf extracts from Osage orange, Filip et al. prepared ethanolic rather than aqueous extracts, and showed that numerous bacterial species, including both *S. aureus* and *E. coli*, could be inhibited by the plant [21]. Our data support these previous findings of the presence of significant antibacterial properties of Osage orange leaf extracts but also suggests that simple water-based preparations of the plant may retain the antibacterial activity found within the leaves of the plant.

A major finding of this study was the antiviral activities of Osage orange and spearmint aqueous leaf extracts against hRSV. Spearmint is widely consumed in tea and other preparations to alleviate colds and respiratory ailments. While other studies have demonstrated that spearmint extract may block or inhibit other viruses, until now, it remained unknown whether spearmint could inhibit hRSV [27,28]. We found that spearmint extract inhibited hRSV infectivity in a dose-dependent manner with an EC_50_ of 1.01 mg/mL, or approximately 1% of the saturated parent leaf extract prepared. Osage orange, while less potent with an EC_50_ of 3.85 mg/mL, was not previously reported to harbor anti-hRSV activity. While carvone, present in spearmint, was shown to significantly inhibit hRSV. The GC-MS analysis of both plant extracts identified the lead compounds present in each plant; however, based on prior chemical analyses, many other major constituents known in other parts of the plants were not identified. This may be due in part to the tissue source, extraction procedure, and/or chemical stability. Furthermore, it should be noted that the chemical analysis of the aqueous plant extracts was performed in a prepared solution for GC-MS which differs considerably in its chemical properties from the parent leaf extract. Consequently, a limitation to the conclusions of this study is that the concentrations of the identified compounds by GC-MS may vary considerably from the actual concentrations in aqueous solution. To address this concern, we tested a wide range of concentrations for each compound; however, we recognize the potential limitations associated with our quantification method. In summary, these data indicate that both Osage orange and spearmint potently inhibit the virus, and further study may be warranted to identify their specific active chemical components and mechanisms of inhibition.

In conclusion, we report here that aqueous leaf extracts of Osage orange and spearmint exhibit significant antibacterial and antiviral activities. Further, we demonstrate for the first time that the major human respiratory pathogen hRSV can be inhibited by leaf extracts of both plants. These findings support further study to better understand their mechanism of action against both bacterial and viral pathogens. While we only investigated a single viral pathogen in this study, our findings support additional study to elucidate any additional antiviral activities these two plant extracts may possess.

## Figures and Tables

**Figure 1 pathogens-14-00776-f001:**
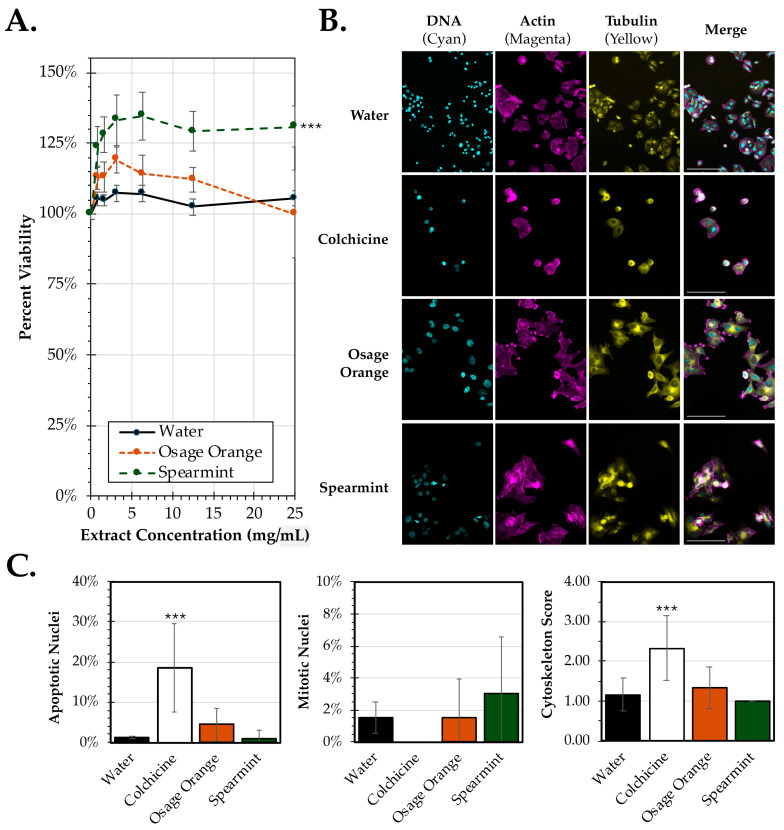
Cell viability and morphology of HEp-2 cancer cells following treatment with leaf extracts from Osage orange (*M. pomifera*) and spearmint (*M. spicata*). (**A**) MTS assay was performed to assess HEp-2 cell viability after 24 h of treatment at 37 °C with extracts or a water control at concentrations up to 25 mg/mL. The percent viability (±SEM) was calculated by comparing activity at each concentration to the 0 mg/mL treatment. (**B**) Representative images of HEp-2 cells after 24 h of exposure to 10 mg/mL of extracts of Osage orange or spearmint, water, or a colchicine control. The cells were fixed and stained with DAPI (DNA; cyan), TRITC-Phalloidin (actin; magenta), and FITC-conjugated anti-tubulin (tubulin; yellow). Scale bar = 100 μm. (**C**) Quantification of the nuclear morphology (left, middle) or cytoskeletal disruption (right) from six images from each treatment. The average percent apoptotic and mitotic nuclei (±SEM) and average cytoskeleton score (±SD) based on blinded scoring from 1 (normal) to 4 (complete cytoskeletal destruction). An ANCOVA was conducted to evaluate significance for the cell viability assay and a one-way ANOVA was used to evaluate the IF scoring. Significant differences relative to water controls are indicated (***, *p* < 0.001).

**Figure 2 pathogens-14-00776-f002:**
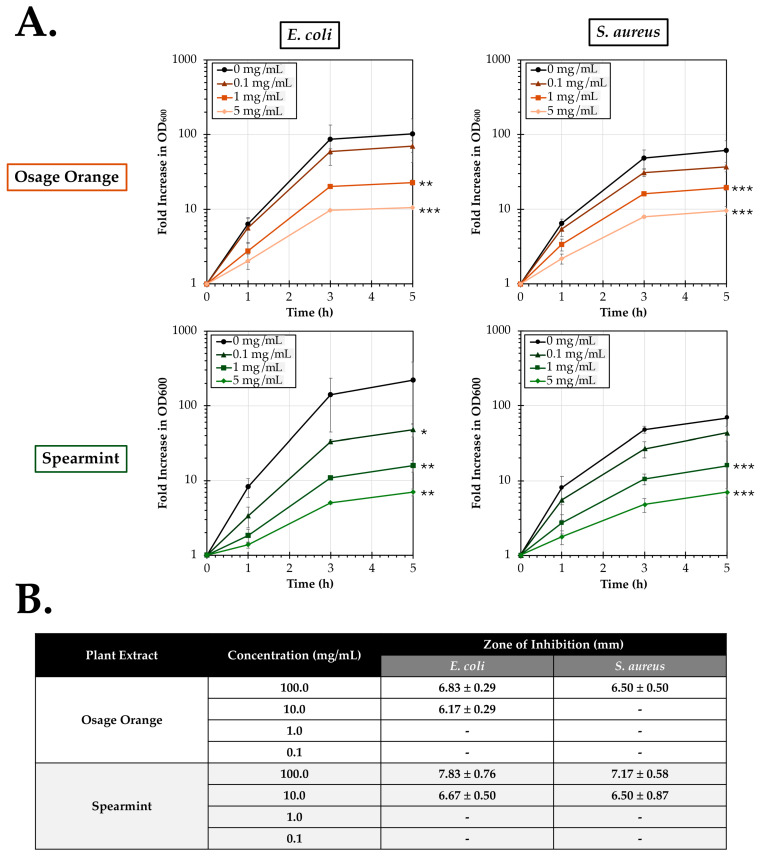
Antibacterial activity of Osage orange (*M. pomifera*) and spearmint (*M. spicata*) extracts. (**A**) Cultures of tryptic soy broth (TSB) were co-inoculated with *E. coli* (left) or *S. aureus* (right) in the presence of doses of Osage orange or spearmint extracts ranging from 0 to 5 mg/mL. Bacterial growth was assessed following growth at 37 °C through measurement of the optical density at 600 nm at 0, 1, 3, and 5 h post-inoculation. The average fold increase (±SEM) in OD_600_ of three experimental replicates relative to the 0 h absorbance is shown. An ANCOVA was used to evaluate significance relative to the untreated 0 mg/mL control. Significant differences are indicated (*, *p* < 0.05; **, *p* < 0.01; ***, *p* < 0.001). (**B**) Disk diffusion tests were performed using 6 mm disks infused with varying concentrations of Osage Orange or spearmint overlaid on cultures of *E. coli* or *S. aureus* on Mueller–Hinton agar (MHA) plates. The average zone of inhibition (±SD; N = 3) was measured after growth at 37 °C for 24 h.

**Figure 3 pathogens-14-00776-f003:**
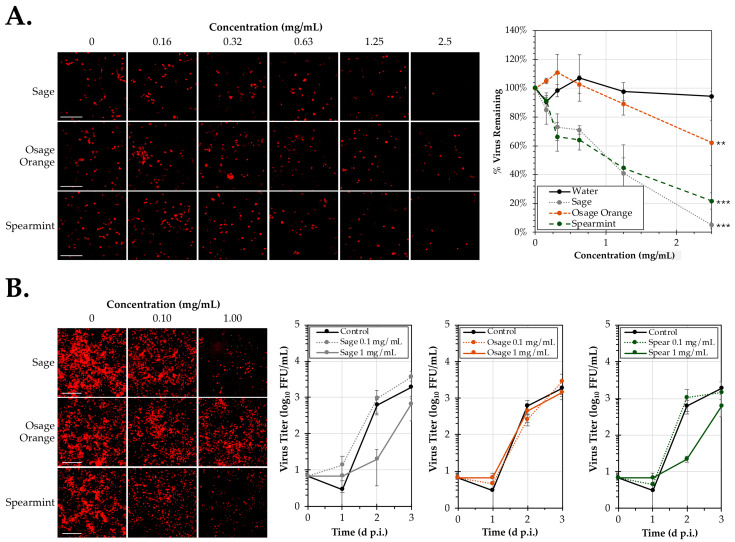
Inhibition of hRSV in Osage orange (*M. pomifera*) and spearmint (*M. spicata*) extracts. (**A**) HEp-2 cells were infected for 24 h with a recombinant fluorescent reporter strain (A2-mKate2) of respiratory syncytial virus (hRSV) in the presence of doses ranging from 0 to 2.5 mg/mL of Osage orange, spearmint, water, or a sage extract control. Red fluorescent foci were visualized (left) and quantified to determine the average percent (±SEM, N = 3) of detected virus remaining relative to the untreated control. (**B**) Replication analysis performed in HEp-2 cells infected at a multiplicity of infection (MOI) of 0.05 FFU/mL over 3 d in the presence of 0, 0.1, or 1 mg/mL of Osage orange, spearmint, or a sage control. The average titers (±SEM; N = 3) of the supernatant are shown as fluorescent focus units (FFU) per mL. Images were taken after 2 d of infection are shown at left. Scale bar = 200 μm. An ANCOVA was used to evaluate overall significance relative to the untreated control. Significant differences are indicated (**, *p* < 0.01; ***, *p* < 0.001).

**Figure 4 pathogens-14-00776-f004:**
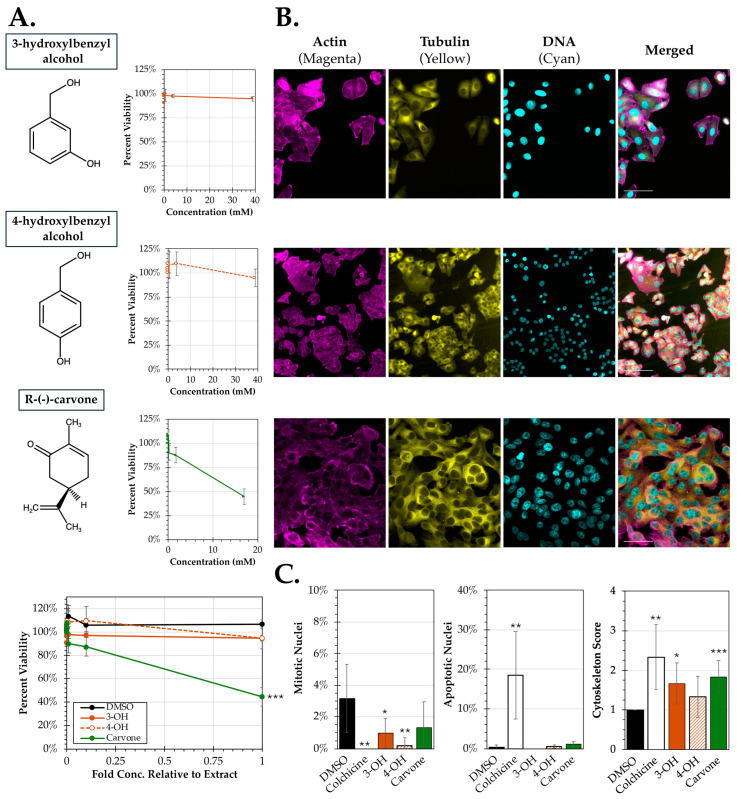
Cell viability and nuclear and cytoskeletal morphology after treatment with chemical constituents identified in Osage orange (*M. pomifera*) and spearmint (*M. spicata*) leaf extracts. (**A**) Chemicals identified by GC-MS screen (left) and their corresponding cell viability determined by an MTS assay performed on HEp-2 cell viability after 24 h of treatment at 37 °C (right). The average percent viability (±SEM) was calculated by comparing activity at each concentration to the 0 mM treatment. (**B**) Representative images of HEp-2 cells after 24 h of exposure at 10% of the concentration in the parent extracts (390 μM 3-hydroxybenzyl alcohol, 390 μM 3-hydroxybenzyl alcohol, and 170 μM carvone). The cells were fixed and stained with DAPI (DNA; cyan), TRITC-Phalloidin (actin; magenta), and FITC-conjugated anti-tubulin (tubulin; yellow). Scale bar = 100 μm. (**C**) Quantification of the nuclear morphology (left, middle) or cytoskeletal disruption (right) from six images from each treatment. The average percent apoptotic and mitotic nuclei (±SEM) and average cytoskeleton score (±SD) based on blinded scoring from 1 (normal) to 4 (complete cytoskeletal destruction). An ANCOVA was conducted to evaluate significance for the cell viability assay and a one-way ANOVA was used to evaluate the IF scoring. Significant differences relative to DMSO controls are indicated (*, *p* < 0.05; **, *p* < 0.01; ***, *p* < 0.001).

**Figure 5 pathogens-14-00776-f005:**
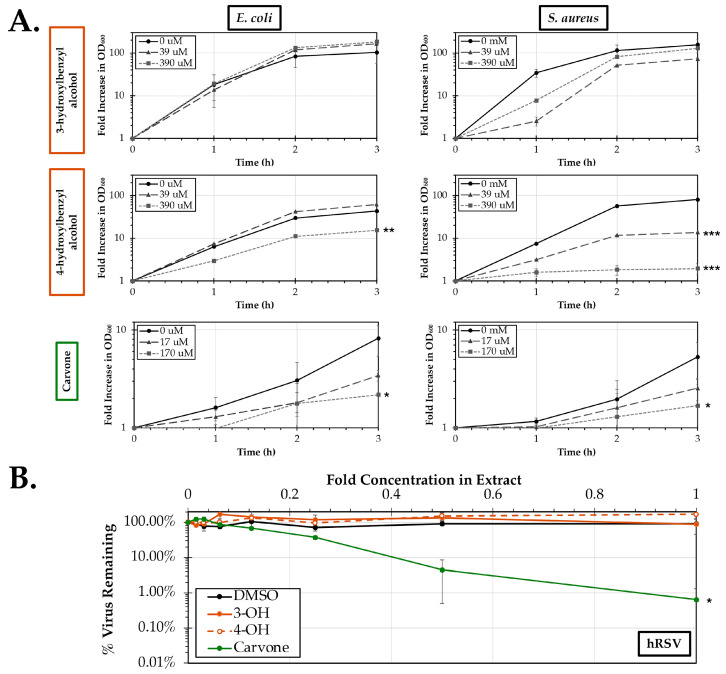
Antibacterial and antiviral activity of chemical constituents derived from Osage orange (*M. pomifera*) and spearmint (*M. spicata*) leaf extracts. (**A**) Cultures of TSB were co-inoculated with *E. coli* (left) or *S. aureus* (right) in the presence of the indicated doses of 3- or 4-hydroxylbenzyl alcohol or R-(-)-carvone. Bacterial growth was assessed through measurement of the optical density at 600 nm. The average fold increase (±SEM) in OD_600_ of three experimental replicates relative to the 0 h absorbance is shown. (**B**) HEp-2 cells were infected for 24 h with a recombinant fluorescent reporter strain (A2-mKate2) of respiratory syncytial virus (hRSV) in the presence of doses of the chemical constituents or a DMSO control. The average percent (±SEM, N = 3) of detected virus remaining relative to the untreated control is shown for all doses relative to their respective concentrations in their parent Osage orange and spearmint extracts. An ANCOVA was used to evaluate significance relative for both antibacterial and antiviral assays to the untreated 0 mg/mL control. Significant differences are indicated (*, *p* < 0.05; **, *p* < 0.01; ***, *p* < 0.001).

## Data Availability

All data and procedures described herein are available upon request.

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
