# Peer review of "Osage Orange (Maclura pomifera) and Spearmint (Mentha spicata) Leaf Extracts Exhibit Antibacterial Activity and Inhibit Human Respiratory Syncytial Virus (hRSV)"

_pathogens, 2025, doi:10.3390/pathogens14080776_

Round 1

Reviewer 1 Report

Comments and Suggestions for Authors

Respiratory syncytial virus (RSV) is one of the main causative agents of acute respiratory infections in children and the elderly. Every year, it causes millions of cases of disease, including severe bronchiolitis and pneumonia, especially in infants and immunocompromised individuals. Despite significant medical and economic costs, there are still no specific antiviral drugs against RSV.

In this regard, the search for and development of effective antiviral drugs is an urgent task of medicinal chemistry and pharmacology.

The presented study is performed at a high level and can be accepted for publication after eliminating a number of comments.

In the section "Materials and Methods" on page 3, lines 9-91, it is necessary to describe in more detail the method for obtaining the extract of the natural compound, indicating the yield of the extract, the duration of extraction with hot water, etc.

In addition, there is practically no description of the GC/MS analysis procedure. It is extremely important to indicate the content of the components identified by the authors.

In addition, the authors refer to article 31, which is not directly related to the study.

An important point is the following: the primary screening uses an aqueous extract, from which further extraction with an organic solvent is carried out. It is necessary to recalculate the content of natural compounds in the aqueous extract, since there is no direct correlation between the solubility of the target compounds in an organic solvent and their presence in the aqueous extract.

In this regard, a more detailed justification for the choice of benzyl alcohols and R-carvone as a study object is required.

EC50 data for the activity of extracts and individual compounds should be presented with an error. In addition, it is advisable to convert the activity values into micromolar units and compare them with previously published data on other viruses, if any.

After making these corrections, the manuscript will be ready for acceptance and publication.

Author Response

Comment: In the section "Materials and Methods" on page 3, lines 9-91, it is necessary to describe in more detail the method for obtaining the extract of the natural compound, indicating the yield of the extract, the duration of extraction with hot water, etc.

ResponseThank you for asking for more clarification on our extract preparation procedure. We have provided more detail pertaining to both preparation of the leaves for extraction as well as the process of making the aqueous extract itself (see Methods, lines 89 - 96).

Comment: In addition, there is practically no description of the GC/MS analysis procedure. It is extremely important to indicate the content of the components identified by the authors.

Response: We have now provided a much more detailed procedure on the GC/MS analysis procedure (see Methods, lines 104 - 114).   

Comment: In addition, the authors refer to article 31, which is not directly related to the study.

Response: Sorry for the confusion. The GC-MS extraction and analysis procedure was originally modified from reference 31.

Comment: An important point is the following: the primary screening uses an aqueous extract, from which further extraction with an organic solvent is carried out. It is necessary to recalculate the content of natural compounds in the aqueous extract, since there is no direct correlation between the solubility of the target compounds in an organic solvent and their presence in the aqueous extract.

Response: This is certainly a limitation of our study that we have now reported in the discussion section in detail (see Discussion, lines 476 – 487). In performing the compound analyses in Figures 4 and 5, we intentionally used a wide range of concentrations in an attempt to capture both the effective concentration in the aqueous extract and any biological activity that may it have. However, identifying the actual concentrations of these compounds in the parent aqueous extracts would be helpful in future study of their impacts.

Comment: EC50 data for the activity of extracts and individual compounds should be presented with an error. In addition, it is advisable to convert the activity values into micromolar units and compare them with previously published data on other viruses, if any.

Response: In our previous submission, we reported the EC50 calculated from the average line of the replicates. We have now recalculated the EC50 values by measuring the value for each experimental replicate and reported the average and standard deviation (see Results lines 291 – 299 and lines 412 - 413). While this did not dramatically change the averages, we have now incorporated error along with our reported EC50 values. We could not find comparable published data for aqueous leaf extract activity for comparison with hRSV.

Reviewer 2 Report

Comments and Suggestions for Authors
  1. Figure 3 missing scale bar.
  2. It will be good to have more supporting assays to show the antibacterial and antiviral activities of the tested extracts. More experimental details should be included in the methodology for these sections. 
  3. Figure 4 legend and the description of the assay is missing details on the exact concentrations use to test and obtain the images. Additionally, what is the objective of performing the morphology studies on the actin, exoskeleton, as well as the nuclear condensation when the emphasis of this article seems to be on the antibacterial and antiviral effects.
  4. What are the EC50 for each of the biological assays conducted for this study? This is not clear.
  5. Units should be correct and standardized. mg/ml should be mg/mL.

Author Response

Comment: Figure 3 missing scale bar.

Response: We have provided scale bars for both infection experiments and have amended the figure legend to reflect the size of the scale bars (see Figure 3).  

Comment: It will be good to have more supporting assays to show the antibacterial and antiviral activities of the tested extracts. More experimental details should be included in the methodology for these sections. 

Response: Thank you for this suggestion and we have conducted two additional experiments – a bacterial disk-diffusion experiment (see Figure 2B) and a viral growth curve experiment (see Figure 3B) – to provide additional insight into the antibacterial and antiviral activities of the two plant extracts. These data are described in the results (see lines 250 – 268 and lines 301 - 325).

Comment: Figure 4 legend and the description of the assay is missing details on the exact concentrations use to test and obtain the images. Additionally, what is the objective of performing the morphology studies on the actin, exoskeleton, as well as the nuclear condensation when the emphasis of this article seems to be on the antibacterial and antiviral effects.

Response: Figure 4 has been updated with the concentrations of extracts used for the immunofluorescence experiment. The purpose of evaluating the cells with immunofluorescence was to ensure that any impacts on hRSV were the result of inhibition of the virus rather than impacts on cell structure or function. We have provided more justification for these experiments in the results (see lines 284 - 288) and discussion (see lines 432 - 439).

Comment: What are the EC50 for each of the biological assays conducted for this study? This is not clear.

Response: A more thorough explanation for what the EC50 is has been provided in the Method section (see lines 167 - 168).

Comment: Units should be correct and standardized. mg/ml should be mg/mL

Response: Thank you for catching the inconsistencies in units here. We have corrected the units for consistency throughout.  

Reviewer 3 Report

Comments and Suggestions for Authors

Major comments

This manuscript describes the findings that the leaf extracts of Osage orange and spearmint have activities against bacterial infection (tested E. coli and S. aureus) and viral infection (tested hRSV) in a cell-based assay, which may have the therapeutic potential for clinic use. The experiments were conducted in the functional and cytotoxicity assays of these leaf extracts. Especially, three constituents were isolated from the leaf extracts and evaluated individually.

Minor comments:

  1. Overall, antiviral activity from both extracts of Spearmint and Osage orange are very weak, the EC50s are over 1mg/ml, even the only purified compound antiviral compound Carvone (Ec50 2.8mM).
  2. Interestingly, the Sage control had very good activity(Figure 3), much better than Osage orange. The detail information such as the Scientific name should be included for this Sage.
  3. Any other viruses were evaluated?

Author Response

Comment: Overall, antiviral activity from both extracts of Spearmint and Osage orange are very weak, the EC50s are over 1mg/ml, even the only purified compound antiviral compound Carvone (Ec50 2.8mM).

Response: While we agree that these concentrations are necessary to observe significant antiviral activity, the significant impacts that we have observed warrant further study to better understand how they are impacting hRSV.  

Comment: Interestingly, the Sage control had very good activity(Figure 3), much better than Osage orange. The detail information such as the Scientific name should be included for this Sage.

Response: We continue to be amazed by the antimicrobial activity of Sage. Sage extract was prepared using the same methodology used for Osage Orange and Spearmint. We have included the full scientific name of Sage and a reference to our previous work with the extract (see lines 288 - 289).

Comment: Any other viruses were evaluated?

Response: Unfortunately, given the scope of this study, we were unable to evaluate other viruses, but would love to do so in the future. We have commented about this point in the discussion (see lines 494 - 497).

Round 2

Reviewer 1 Report

Comments and Suggestions for Authors

The article can be accepted for publication.

Reviewer 2 Report

Comments and Suggestions for Authors

Authors have addressed majority of the comments.